# Wastewater Treatment Costs: A Research Overview through Bibliometric Analysis

Leticia Gallego-Valero [1,*], Encarnación Moral-Parajes [1] and Isabel María Román-Sánchez [2]

1 Department of Economy, University of Jaén, 23071 Jaén, Spain; Emoral@ujaen.es
2 Department of Economics and Business, University of Almería, 04120 Almería, Spain; Iroman@ual.es
* Correspondence: Lgallego@ujaen.es; Tel.: +34-651888733

**Abstract:** Given the problem of water scarcity and the importance of this resource for the sustainability of the planet, wastewater treatment and its costs have become a key issue for proper water management. Using bibliometric analysis of publications in the Web of Science database, this study presents an overview of the research on wastewater treatment costs in the period 1950–2020. The worldwide search returned 22,788 articles for wastewater treatment costs, which compares poorly to the results for research on wastewater treatment, accounting for only 12.34% of the total output on wastewater treatment. The findings of this study reveal the leading countries in this field of research (China, USA, India, Spain and the UK), with the articles being published in a wide range of high impact journals. Similarly, there are very few results on UV and chlorination costs, despite the importance of these two treatments for wastewater disinfection and reuse. This study is aimed at researchers in this field, helping them to identify recent trends, and at the main institutions in the scientific community working on this subject.

**Keywords:** cost; database; treatment; wastewater; water; Web of Science





## 1. Introduction

Given the importance of water resources, appropriate water management is needed as well as more sustainable exploitation of this resource [1–4]. At a global level, water scarcity is an economic, sanitation and even security issue [5]. Moreover, the problem is expected to become more acute in the future, with this resource playing a fundamental role in the sustainability of the planet [6,7]. The United Nations (UN) considers clean water and sanitation to be a priority objective, and one of its goals is to ensure universal access to safe and affordable drinking water. European Union policy is also aimed at protecting this resource, with the implementation of the Water Framework Directive (2000/60/EC) and the Urban Waste Water Treatment Directive (91/271/EEC). Factors such as the population growth in many urban areas, agricultural productivity, the economic development of different countries, industrialization, energy production, improvements in health and sanitation systems, and the expansion of irrigation systems in arid regions, have underscored the fact that conventional resources alone cannot meet the constantly growing demand [8–14].

Wastewater costs suppose a great concern given the need for a growing resource [15]. In response to the problem of scarcity, which has become hugely important in countries with high levels of water stress [16], hydric resources should be managed more efficiently [17]. Appropriate water management aimed at increasing the supply of water necessarily involves the use of wastewater treatment. [18–20]. Wastewater management is expensive and poses problems regarding how to finance it and how to reduce treatment costs [21]. Adequate wastewater management is necessary to finance the investment in wastewater treatment plants (WWTPs) and the costs of treatment technologies, and to improve the environmental quality of water resources [22,23]. Treatment methods improve the quality of the water and, when the treated water is reused, increase the quantity of the resource [24,25].

Water reuse is a process with few adverse environmental impacts when compared with desalination or water transfers and offers economic and social benefits [17]. This process gives rise to a resource, the reused wastewater, that can help to improve the quality and quantity of the planet's water supplies [26–28]. Treating wastewater prior to its discharge helps ensure the good status of water resources [29], facilitating the use of reclaimed water as an additional source of water supply that is safe and economical [30]. In view of the growing demand, sanitation and purification treatments constitute an indispensable tool for cleaning the water that is returned to ecosystems and increasing the quantity of the available resource, regardless of climate conditions [31–34]. This way, reused wastewater can be seen as a source of irrigation supply that is both economical and safe in terms of human health and the environment [35–37], helping to tackle the problem of water scarcity [38–40], boosting supply and decreasing the dependence on groundwater and surface water resources [41].

The analysis and study of the costs of the different treatments is crucial in order to boost their efficiency, cut costs and help ensure the widespread use of such treatments [42]. There is a need for cheaper, more robust and more effective processes for wastewater decontamination and disinfection, always bearing in mind the need to protect human health and the environment [17]. It is increasingly important to adopt appropriate measures to bring down operating costs, which entails an evaluation of the efficiency of WWTPs. By doing so, it is possible to identify WWTPs that make better use of their economic resources without reducing the quality of the treated water. This information can then be used to determine the appropriate operational practices to be applied in other WWTPs in order to reduce operating costs. In addition, this cost-cutting is beneficial to society as a whole, since it is the citizens who bear these costs through the payment of water tariffs [43]. Economic evaluation is also a useful tool in the implementation of efficient and effective water management strategies and policies, thus supporting various institutions' policy decisions [44–46].

The present study is carried out through bibliometrics, a technique that uses statistical methods to analyse the scientific output published and which contains sub-fields such as structural, dynamic, evaluative and predictive scientometrics. Bibliometric analysis has been applied to almost all scientific fields, and all types of literature can be studied in this way, identifying features such as topics, authors, publication dates, reference literature, content, etc. [47,48]. The use of the internet as a data collection tool is accepted by the scientific community [49]. In this regard, Web of Science (WoS), published by Thomson Reuters, is a hugely relevant database for evaluating research [50].

The main aim of this work is the quantitative and qualitative analysis of the dynamics of global research on the costs of wastewater treatments since 1950, as well as an analysis of the research on the costs of chlorination and UV disinfection treatments. These treatments enable the reclamation of water for reuse, which contributes to an efficient management of the resources used, keeping them circulating in the economic system for as long as possible, and thereby generating less waste and avoiding the unnecessary use of new resources. They therefore help to reduce environmental impacts, as well as contributing to the restoration and regeneration of natural capital, in line with the tenets of the circular economy [51]. The application of these treatments contributes to sustainability by allowing the value of resources to remain in the economy for as long as possible and reducing waste generation to a minimum. To achieve this objective, bibliometric techniques are used to identify, organize and analyse the main elements of the topics in question, using the WoS database and statistical processing tools. The results obtained are useful for the scientific community to gain an understanding of the current environment and upcoming trends in the lines of research on these subjects, and to make decisions before embarking on research.

## 2. Materials and Methods

### 2.1. Bibliometric Analysis

The article performs a bibliometric analysis of wastewater treatment costs. This method makes it possible to identify, organize and evaluate the constituent elements of a specific area of study, and is, thus, an important tool for literature reviews [52–54]. Bibliometric analysis involves various different types of indicators relating to quantity (productivity), quality (impact of publications) and structure (analysing connections) [55]. This article uses the WoS database from Thomson Reuters to conduct the bibliometric analysis. WoS has high visibility in the different areas of knowledge, a selection filter for prestigious publications, and is also widely used to carry out bibliometric studies [56,57]. The bibliometric analysis technique has been used to study areas such as the use of water or wastewater [58,59], wastewater treatment by advanced oxidation processes [17], infectious diseases and microbiology [60], or renewable energies, sustainability and the environment [61].

### 2.2. Data Selection and Processing

The sample of documents analysed in this study was obtained by conducting a search of the entire WoS database with the term "wastewater treatment cost" in the option "topic":

- The period analysed was 1950–2020. The analysis yielded a final sample of 22,788 results on wastewater treatment costs. The sample selection was conducted in January 2021.
- The following variables were analysed: evolution, areas of study, main countries, main journals, and main institutions.
- An additional analysis section was introduced to examine differences in scientific research on treatments for water intended for reuse, ultraviolet (UV) and chlorination, as these treatments are the most commonly used options for water disinfection, for various economic and environmental reasons.

Bibliometric studies distinguish between three types of indicators [55]: quantity indicators, which refer to productivity; quality indicators, which refer to the impact of publications; and structural indicators, which measure the connections established between the different agents. In this study, quality and quantity indicators were analysed. In addition to the measure of productivity of the countries and institutions, the following indicators were used to evaluate the quality of the journals in which the documents were published:

- SCImago Journal Rank (SJR): measure of the scientific influence of scholarly journals that accounts for both the number of citations received by a journal and the importance or prestige of the journals where the citations where made [62].
- Quartile in which the journal is positioned.
- Number of citations.
- Journal Impact Factor (JIF): measure of the frequency with which the average article in a journal has been cited in a particular year.
- Total publications.

After selecting the sample, the information available in the WoS database was downloaded and prepared for analysis by eliminating duplications, correcting mistakes and adding non-complete information [54]. First, the evolution of the field over the period 1950–2020 was analysed. Secondly, the main areas of study in wastewater research classified by WoS were identified, before reviewing the leading countries in this research. In the next step, the main journals were identified along with their SJR, JIF and total citations index for the year 2019 (to evaluate the impact of the journals). The main institutions are shown below. Lastly, the results of the search for research on the costs of UV and chlorination treatments were studied.

### 3. Results

*3.1. Evolution of the Research on Wastewater Treatment Costs*

The scientific community has shown very little interest in analysing wastewater treatment costs, although there has been some growth in the theoretical and applied literature on the subject over the last decade. Figure 1 presents the evolution of articles published in this field from 1950 to 2020, revealed by the bibliometric analysis using WoS (2020) as the main database. The search for "wastewater treatment cost" returns 22,788 results, a very small number compared to those returned by the search "wastewater treatment", with 184,697 articles published. In percentage terms, wastewater treatment cost research comprises only 12.34% of the total for wastewater treatment. The research on wastewater treatment costs shows an increasing trend between 1950 and 2020, with a marked rise from 2010, albeit with some fluctuations. It can be seen that, in both cases, the majority of articles are concentrated from 2010 onwards (16,204 studies, 71.11% of the total for "wastewater treatment cost" and 125,235 articles, 67.81% of the total for "wastewater treatment"). The period between 1950 and 1980 yields very few results, although the number grows over time. Looking at the results since 1980, 20 articles were published for "wastewater treatment cost" and 235 for "wastewater treatment" in 1980, while the corresponding figures for 2020 were 2528 and 16,708, respectively. The growth in the research on wastewater treatment costs is slower than that for wastewater treatment: although both show a rising trend during the period 1950–2020, research on "wastewater treatment" grows at a much faster rate, especially after the year 2000.

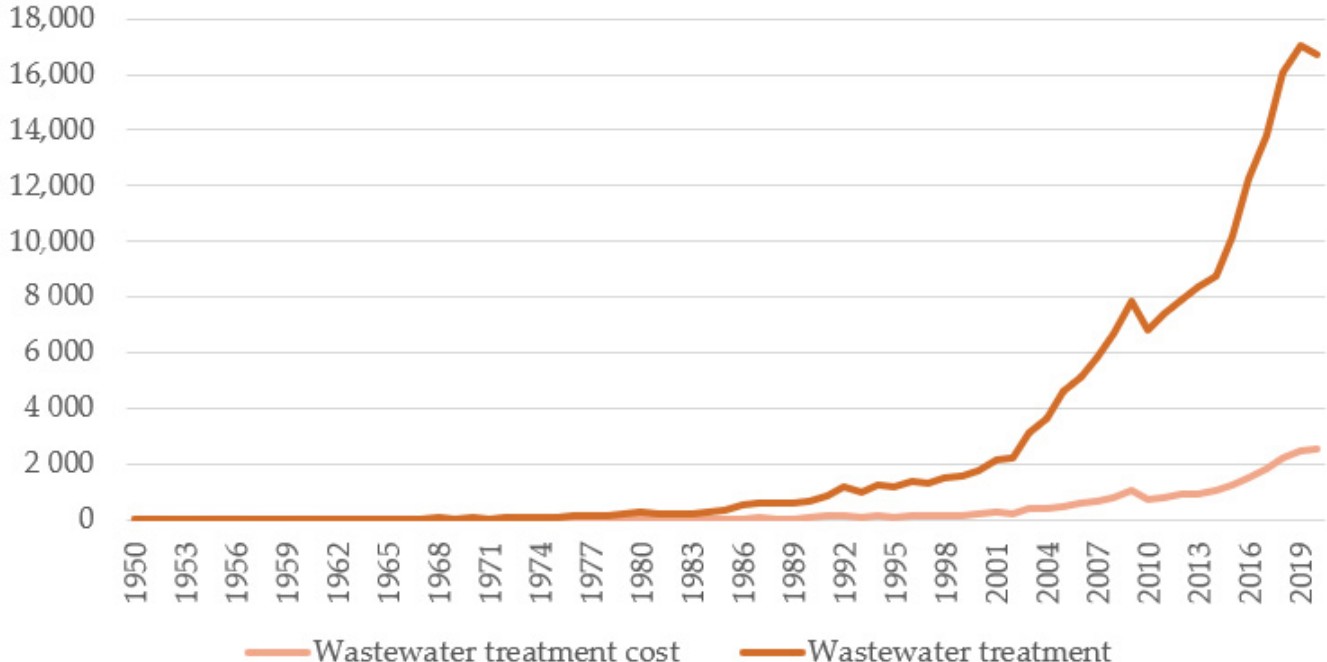

**Figure 1.** Trends in the research on wastewater treatment costs and wastewater treatment (number of articles). Source: own elaboration from WoS (2021).

*3.2. Main Areas of Study in Wastewater Treatment Costs Research*

The results of the bibliometric analysis make it possible to distinguish between the different disciplines to which the analysed scientific articles belong. It should be noted that an article can belong to more than one category; for this reason, the results are analysed in percentages. Figure 2 shows the main areas of study in wastewater treatment costs in the period under study. Among the many areas of research, the most important are Environmental Sciences and Ecology, accounting for 14% of the total, Engineering (11%) and Water Resources (10%). These areas are followed by Public Environmental Occu-

pational Health (8%), Energy and Fuels (6%), Business Economics (6%) and Chemistry (6%). The item labelled "other" (39% of the total) includes a wide and diverse range of areas, none of them with a percentage higher than 5%, such as Materials Science, Toxicology, Biochemistry Molecular Biology, Mathematics, Biodiversity Conservation, Physics, Marine Freshwater Biology, Plant Sciences, Microbiology, Mathematical Computational Biology, Computer Science, Food Science Technology, Polymer Science, Meteorology Atmospheric Sciences, Science Technology (other topics), Biotechnology Applied Microbiology, Instruments Instrumentation and Agriculture.

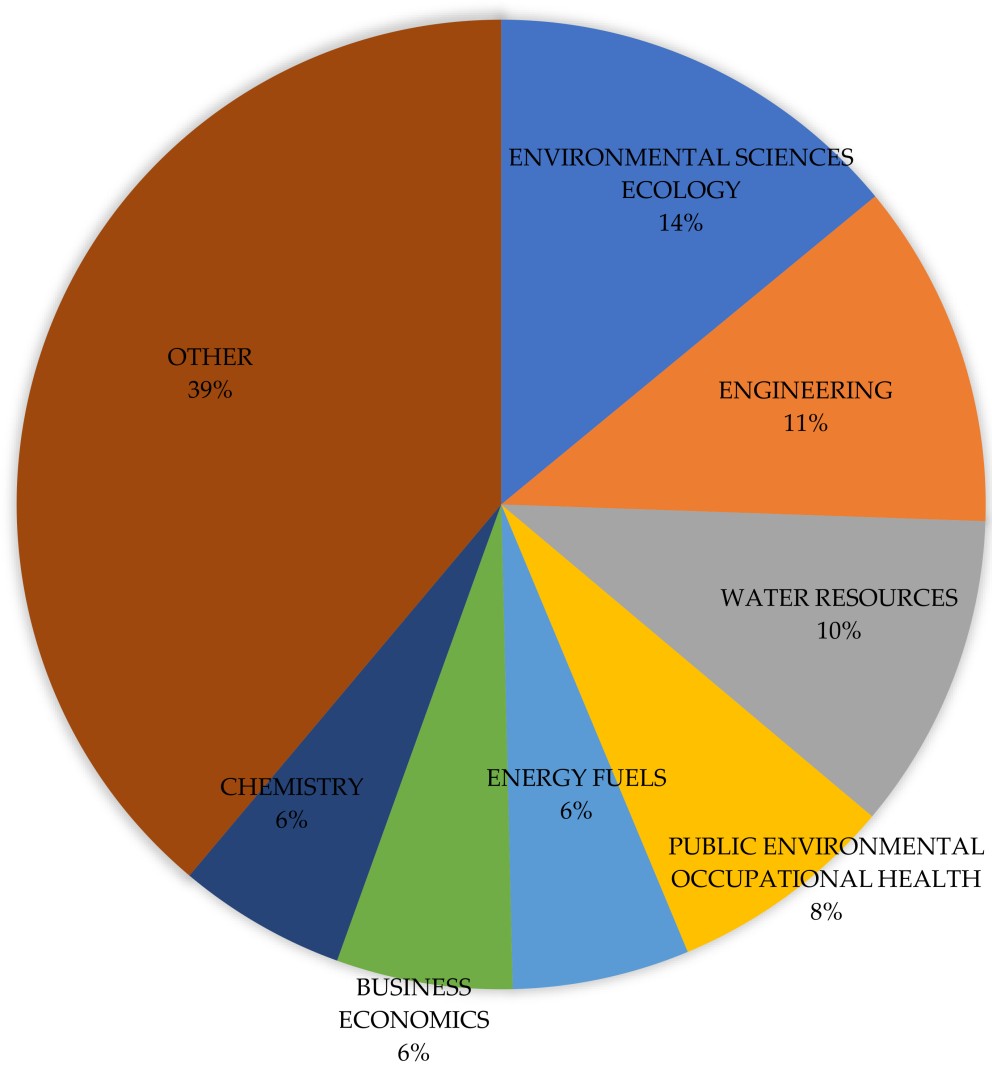

**Figure 2.** Main areas of study in wastewater treatment costs (percentage). Source: own elaboration from WoS (2021).

### 3.3. Relevant Countries in Wastewater Treatment Costs Research

The articles published on wastewater treatment costs in the period 1950–2020 come from a total of 162 countries. Figure 3 shows the map of the countries with results on wastewater treatment cost research, although the majority of studies in this field come from a relatively small number of main countries. In order of quantity of results, they are China, the USA, India, Spain, the UK, Australia, Brazil, Canada, Turkey and Iran. Altogether, there are countries from almost all continents, and with a wide diversity of economic and sociocultural characteristics.

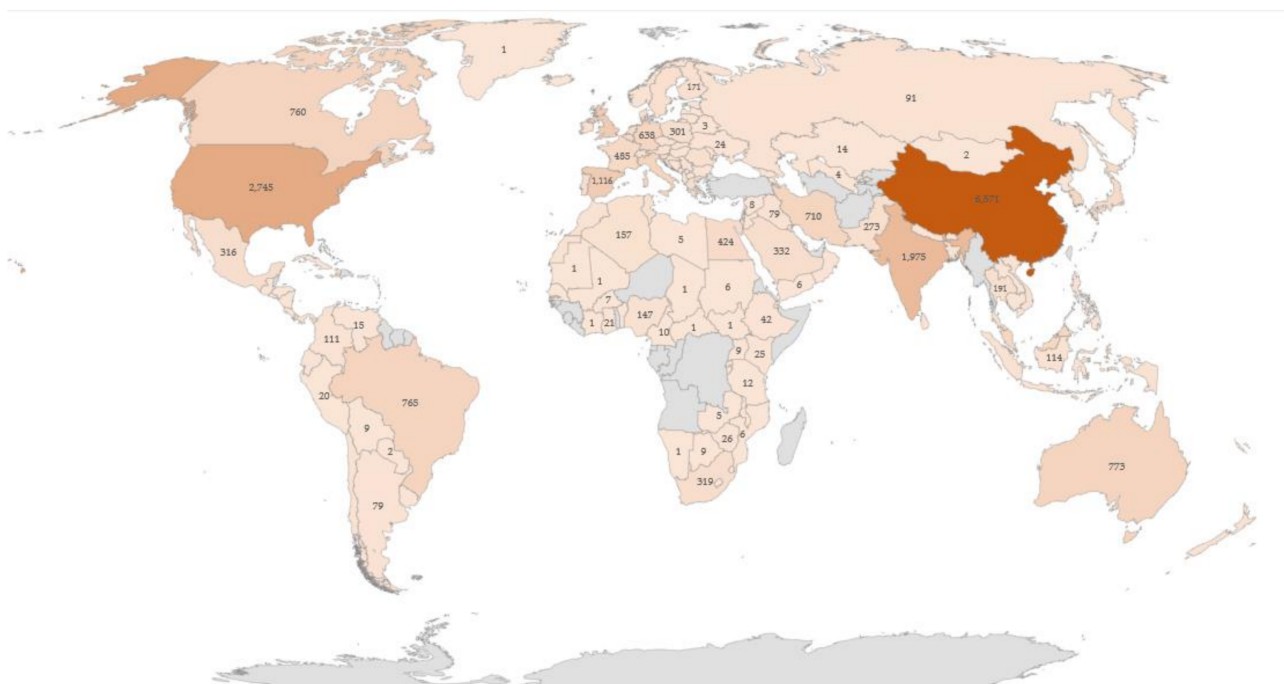

**Figure 3.** Map of the research on wastewater treatment costs (number of articles). Source: own elaboration from WoS (2021).

Figure 4 presents the top countries in the field. China is the most prolific country, representing 28.84% of the total, with 6571 results. The second place is occupied by the USA (12.05% of the total, 2745 articles). Third is India, with 1975 results and 8.67%. These countries are followed by Spain (1116 results, 4.90%), the UK (1112 results, 4.88%), Australia, Brazil, Canada, Turkey, Iran, Italy, Germany, Malaysia, South Korea, France, Japan, Egypt and the Netherlands.

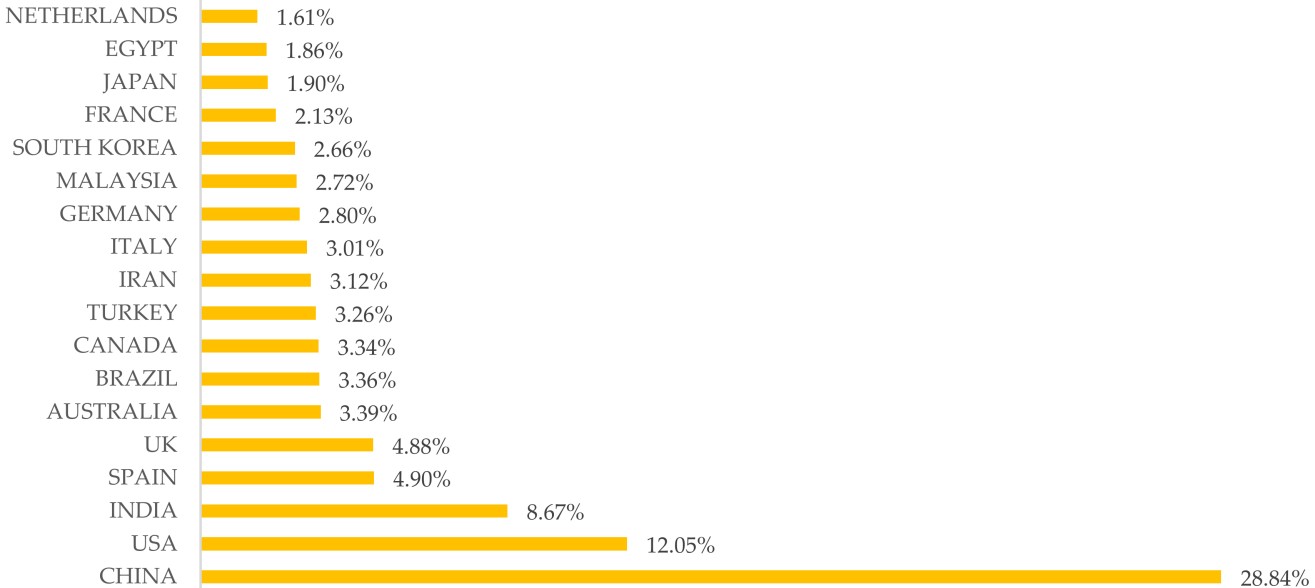

**Figure 4.** Top countries in wastewater treatment cost research (percentage). Source: own elaboration from WoS (2021).

### 3.4. Journals in Wastewater Treatment Cost Research

This part presents the most relevant journals publishing articles on wastewater treatment cost research and analyses their main indexes (Table 1). The main journals by number of articles are Water Science and Technology (1471 articles), Chemical Engineering Journal (709) and Desalination and Water Treatment (651). Taken together, these three journals published 12.42% of the total papers on this research subject. Among the list of journals, there are diverse nationalities, with the United Kingdom, the Netherlands, Switzerland and the USA being the most prolific countries.

**Table 1.** Journals and relevant indexes in wastewater treatment cost research. Source: own elaboration based on WoS (2021).

| Journal | Articles | SJR (2019) | Country | JIF (2019) | Total Citations (2019) |
|---|---|---|---|---|---|
| Water Science and Technology | 1471 | 0.47 (Q2) | United Kingdom | 1.638 | 20,937 |
| Chemical Engineering Journal | 709 | 2.32 (Q1) | Switzerland | 10.652 | 129,806 |
| Desalination and Water Treatment | 651 | 0.33 (Q2) | Italy | 0.854 | 14,535 |
| Journal of Hazardous Materials | 552 | 2.01 (Q1) | Netherlands | 9.038 | 110,068 |
| Water Research | 523 | 2.93 (Q1) | United Kingdom | 9.130 | 99,442 |
| Journal of Cleaner Production | 492 | 1.89 (Q1) | Netherlands | 7.246 | 104,138 |
| Bioresource Technology | 468 | 2.43 (Q1) | Netherlands | 7.539 | 131,781 |
| Journal of Environmental Management | 452 | 1.32 (Q1) | USA | 5.647 | 44,264 |
| Environmental Science and Pollution Research | 415 | 0.79 (Q2) | Germany | 3.056 | 46,033 |
| Desalination | 320 | 1.81 (Q1) | Netherlands | 7.098 | 44,845 |
| Environmental Technology | 296 | 0.49 (Q2) | United Kingdom | 2.213 | 7947 |
| Journal of Environmental Chemical Engineering | 281 | 0.93 (Q1) | United Kingdom | 4.300 | 13,023 |
| Science of the Total Environment | 254 | 1.66 (Q1) | Netherlands | 6.551 | 134,962 |
| Chemosphere | 240 | 1.53 (Q1) | United Kingdom | 5.778 | 94,799 |
| Environmental Science & Technology | 221 | 2.7 (Q1) | USA | 7.864 | 187,995 |
| Journal of Chemical Technology and Biotechnology | 202 | 0.66 (Q1) | United Kingdom | 2.750 | 12,232 |
| Water | 176 | 0.66 (Q1) | Switzerland | 2.544 | 13,460 |
| International Journal of Environmental Science and Technology | 174 | 0.52 (Q2) | USA | 2.540 | 6522 |
| Water Environment Research | 173 | 0.3 (Q3) | USA | 1.369 | 3120 |
| Water, Air and Soil Pollution | 169 | 0.54 (Q2) | Switzerland | 1.900 | 15,219 |

It is important to note that there is a broad range of journals publishing articles on wastewater treatment costs. The journal with the largest number of articles is Water Science and Technology, with 6.46% of the total sample. This journal has an SJR in 2019 of 0.47 (quartile Q2), a JIF of 1.638 and a total of 20,937 citations in 2019. The second is Chemical Engineering Journal, with 3.11% of the total sample. The SJR in 2019 for this journal is 2.32 (Q1), with a JIF of 10.652 in 2019 and 129,806 total citations in 2019. In third place is Desalination and Water Treatment, with 2.86% of the total sample and an SJR (2019) of 0.33 (Q2), a JIF index of 0.854 and 14,535 total citations. Below these three results, there is a wide variety of journals, almost all of which present high index scores—most are in the first and second quartiles. These 20 journals published 36.15% of the articles, most of them included in the first two quartiles of the SJR. They include Journal of Hazardous Materials, Water Research, Journal of Cleaner Production, Bioresource Technology, Journal of Environmental Management, Environmental Science and Pollution Research, Desalination, Environmental Technology, Journal of Environmental Chemical Engineering, Science of the Total Environment, Chemosphere, Environmental Science & Technology, Journal of Chemical Technology and Biotechnology, Water, International

Journal of Environmental Science and Technology, Water Environment Research, and Water, Air and Soil Pollution.

### 3.5. Leader Institutions in Wastewater Treatment Cost Research

Figure 5 presents the main institutions focusing on wastewater treatment costs research. This large number of institutions (62) accounts for 31.10% of the total results, which indicates a low concentration index at the institutional level. The leading institution in number of related articles is the Chinese Academy of Sciences (China), with 561 articles (2.46% of the total), followed by the Indian Institute of Technology System (India), with 445 articles (1.95% of the total). Institutions from China, India and the USA predominate, with those from the USA having a lower concentration by institution. It is worth noting the wide dispersion of articles among different institutions.

### 3.6. Differences in Scientific Research on Treatments for Wastewater Intended for Reuse: UV and Chlorination

This section analyses the results on disinfection treatments for wastewater intended for reuse; namely, UV and chlorination. These are the most commonly applied treatments for wastewater disinfection for various economic and environmental reasons [63]. The classification as UV or chlorination reveals a difference, with 1118 articles for UV and 140 for chlorination, representing 4.91% and 0.61%, respectively, of the total research on wastewater treatment costs. This is a very small number of results considering the importance of these treatments for the reuse of wastewater, and the fact that they are the most commonly used options. The analysis of the number of publications over the period 1950–2020 shows a growing amount of research on wastewater treatment costs relating to UV and chlorination over the years. Figure 6 shows the evolution of the studies published on these two treatments throughout the period analysed, clearly depicting the scant research up until the year 2000. From then on, we see a rising trend in both cases, with a particularly notable increase in studies on UV, especially from 2014 onwards. In the last 10 years, we see a rise in publications on UV from 29 in 2010 to 147 in 2020, representing a growth rate of 36.99%. In parallel, those addressing chlorination register an increase of 27.27% in the last decade, with two results in 2010 and eight in 2020.

Figure 7 shows the leading countries in research on wastewater treatment costs related to UV and chlorination: China is the leader on UV treatment, with 256 results (22.90% of total results for UV), and the USA for chlorination treatment, with 31 articles (22.14% of total results for chlorination). In the research on the costs of tertiary treatment with UV, China is followed by India, the USA, Spain, Iran, Brazil, Italy, Canada, Germany and Egypt, in that order. In the case of chlorination cost research, after the USA come China, Brazil, India, Australia, Canada, France, Italy, Spain and the United Kingdom. It can be seen that 8 of the 10 leading countries are the same for the two treatments, although they hold different positions according to their numbers of results.

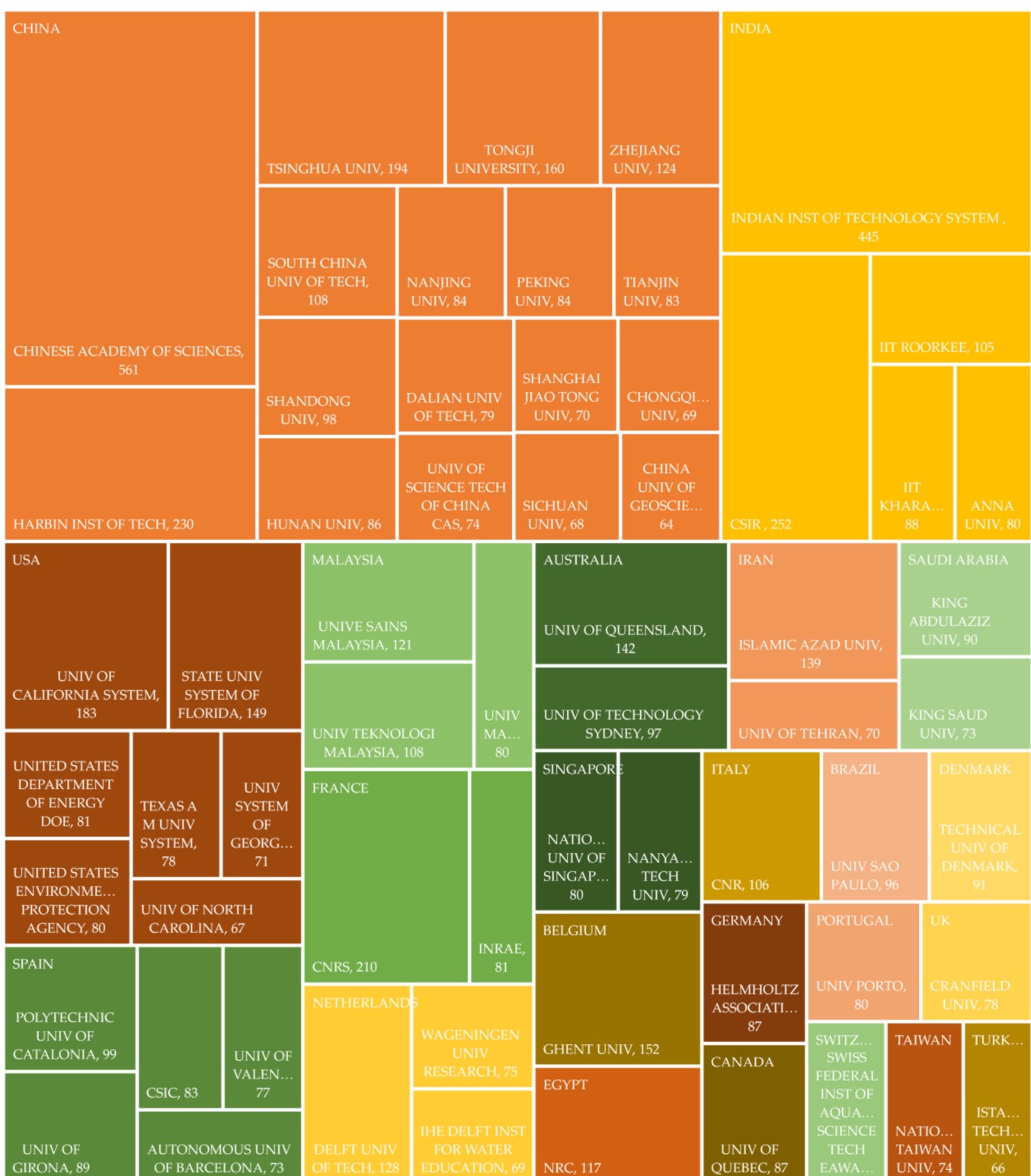

**Figure 5.** Main institutions in wastewater treatment cost research. Source: own elaboration from WoS (2021).

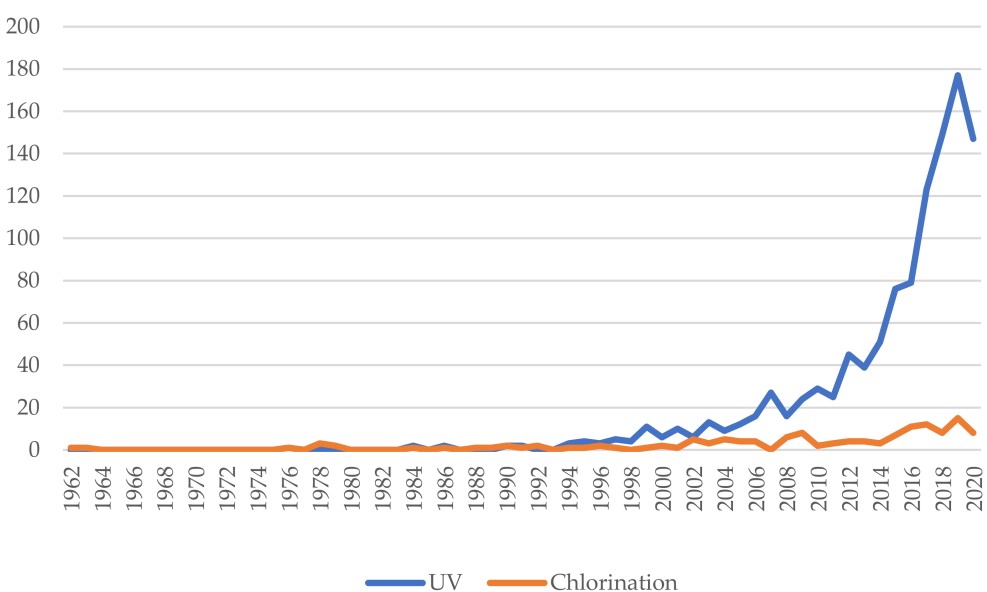

**Figure 6.** Evolution of articles on UV and chlorination treatments for wastewater reuse from 1950 to 2020 (number of articles). Source: own elaboration from WoS (2021).

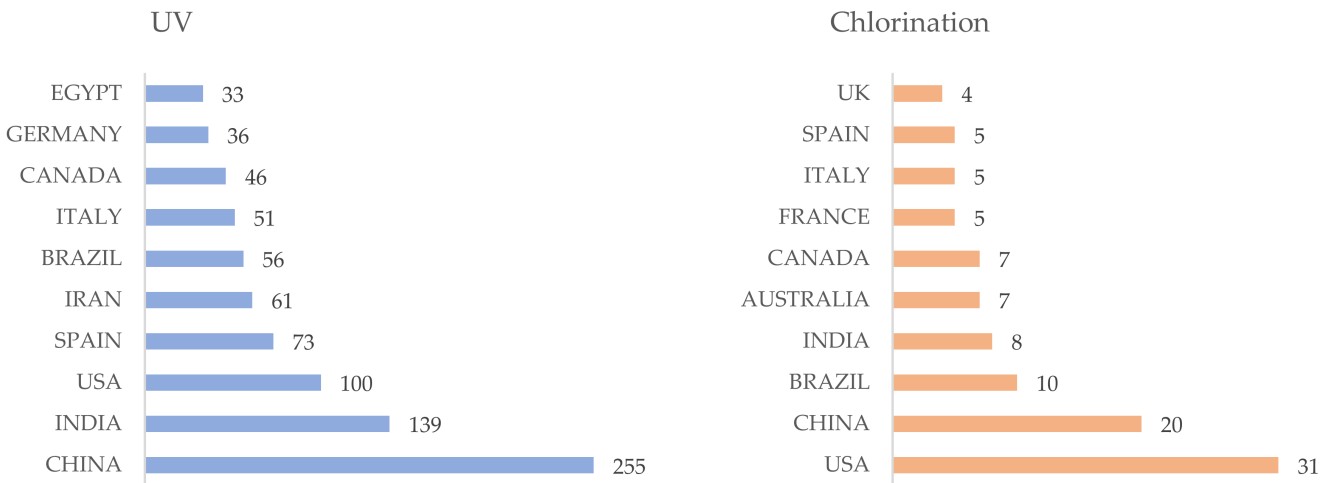

**Figure 7.** Leading countries in the research on wastewater treatment costs related to UV and chlorination (number of articles). Source: own elaboration from WoS (2021).

## 4. Discussion and Conclusions

### 4.1. Summary of Findings

The objective of this work was to show the current status and evolution of research on the wastewater treatment costs during the period 1950–2020. To achieve this goal, the main drivers of the subject, the main lines of research, the trends over several years, and the gaps in research were analyzed in depth. In addition, two disinfection treatments for water reuse have been considered: chlorination and UV. Bibliometric techniques have been used to carry out this study, with the WoS database and statistical processing tools. Globally, research on wastewater treatment costs has yielded 22,788 results: an increasing trend of wastewater treatment costs research was observed. Nevertheless, this topic remains largely understudied compared to the results for the search "wastewater treatments", representing only 12.34% of the total output for research on wastewater treatments. For the sake of easy understanding, the findings of this study are summarized as follows:

(1) The evolution of research suggests that the study of wastewater treatment costs has experienced a steady increase from 2010. This changing trend is related to the growing social concern for the environment and its resources. Reflecting the rise in "wastewater treatment" research, there has been a rise in research on wastewater treatment costs since 1950, showing strong growth since 2010, with 71.11% of the articles being published from this date onwards.

(2) The main areas of research on this topic are very diverse, with the most important being Environmental Sciences and Ecology (14% of the total), Engineering (11%) and Water Resources (10%).

(3) As for the countries of origin of the scientific output in this field during the period 1950–2020, it is worth noting the high dispersion; all together, 162 countries were responsible for the results in wastewater treatment costs research. The main countries are China (28.84% of the total), the USA (12.05%), India (8.67%), Spain (4.90%) and the UK (4.88%).

(4) Concerning to the leader institutions, there is great diversity from which the research on this subject comes, as reflected in the Low Concentration Index at the institutional level; some that stand out are the Chinese Academy of Sciences (China), with 2.46% of the total, and the Indian Institute of Technology System (India), with 1.95% of the total.

(5) Journal analysis revealed that the articles relating to the analysed subject are published in international journals with a high impact factor. There is a wide range of journals that publish articles on the costs of wastewater treatment. Researchers' preferred journals are Water Science and Technology, Chemical Engineering Journal, and Desalination and Water Treatment, with high impact factors of 1.638, 10.652 and 0.854 (JIF), respectively. Taken together with the number of publications, these are an indicator of the high level of institutional scientific quality. These three journals together have published 12.42% of the total research in this field.

(6) Regarding to UV and chlorination costs, very few results were returned in the search for articles (4.91% for UV and 0.61% for chlorination) in comparison with the total results for wastewater treatment costs. It is remarkable that these topics have registered an increase over the period analysed, with a particularly notable rise from the year 2000. The leading countries in this research are the following:

- For UV, China (22.90%), India, the USA, Spain, Iran, Brazil, Italy, Canada, Germany and Egypt.
- For chlorination, the USA (22.14%), China, Brazil, India, Australia, Canada, France, Italy, Spain and the United Kingdom.

*4.2. Implications and Limitations*

The bibliometric analysis conducted in this study shows the trends in wastewater treatment costs research. It is very scarce in comparison with the research on wastewater treatment, highlighting the existence of a research field yet to be explored. The findings of this study provide valuable information for researchers and institutions, helping them explore much-needed management pathways. In practical terms, this study focuses on the importance of the improvements in the management of wastewater treatment.

Despite its contributions, this study has limitations. It is relevant to remark the fact that the findings of this study might not fully reflect the complete research on wastewater treatment costs, given that the information is derived only from the WoS database. There are other databases, such as Scopus, with quality publications, that are not taken into account.

*4.3. Future Research Opportunities*

Using a bibliometric analysis and the WoS, this study identifies the contributions to the topic of wastewater treatment costs, with China, the USA, India, Spain and the UK being the leading countries in this regard. From 2010, we see greater concern for sustainability, which has been promoted by the UN since 2000 when it declared its Millennium Development Goals. Indeed, one of those goals was to ensure environmental sustainability, which

was followed by the inclusion of water in the Sustainable Development Goals in 2015. Nevertheless, the bibliometric analysis carried out reveals a topic that calls for further research, given the scarcity of scientific publications in the area of wastewater treatment costs. In addition, the few results obtained for UV and chlorination costs contrast to the prominence of these two treatments in the disinfection and reuse of wastewater; indeed, they are the most commonly used options for this purpose. The scientific contribution in this field is minimal compared to that on wastewater treatments, revealing a segment that has yet to be explored. This study serves as a guide for researchers, pointing to new trends, and also informs the scientific community and institutions, while the growing interest in the subject enables improvements in the management of wastewater treatment. In addition, society should be more aware of the need to fund wastewater treatments given the relevant role they play in securing additional water resources.

Wastewater reclamation has the disadvantage of entailing high costs, above the average of those associated with naturally-occurring resources. The study of wastewater treatment costs is essential for the proper management of water resources. The decision-making process regarding the planning and management of water resources requires the constant generation of information to achieve high levels of efficiency. Increasingly, the use of reclaimed water has strategic value in that it takes the pressure off water resources, while minimizing health risks for downstream users and helping to maintain the quality of ecosystems. Wastewater treatment and reclamation processes are an essential element of the efficient management of the water cycle, becoming hugely important in countries with high levels of water stress.

**Author Contributions:** Conceptualization, L.G.-V., E.M.-P. and I.M.R.-S.; methodology, L.G.-V. and I.M.R.-S.; validation, E.M.-P. and I.M.R.-S.; formal analysis, L.G.-V.; investigation, L.G.-V.; resources, L.G.-V.; data curation, L.G.-V.; writing—original draft preparation, L.G.-V.; writing—review and editing, E.M.-P. and I.M.R.-S.; visualization, L.G.-V.; supervision, E.M.-P.; project administration, E.M.-P. All authors have read and agreed to the published version of the manuscript.

**Funding:** This research received no external funding.

**Institutional Review Board Statement:** Not applicable.

**Informed Consent Statement:** Not applicable.

**Data Availability Statement:** The data presented in this study are available on request from the corresponding author.

**Conflicts of Interest:** The authors declare no conflict of interest.

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
