# Peer review of "Wastewater Treatment Costs: A Research Overview through Bibliometric Analysis"

_sustainability, doi:10.3390/su13095066_

Round 1

Reviewer 1 Report

Using the bibliometric analysis of publications in the Web of Science database, the authors of this work prepared a transparent study of research and published works in the period 1950 - 2020 with a focus on wastewater treatment for reuse. It is clear that higher quality of treated effluent wastewater is associated with higher costs. An important fact and at the same time a disadvantage of such wastewater regeneration is that they are accompanied by high costs that exceed the average costs associated with the direct use of water from natural sources. This aspect increases the importance of this study. It is a useful material that summarizes research and publication activities in the field of wastewater treatment costs for the purpose of their reuse. Given the increases in water consumption as well as the impact of climate change on water resources, it is a realistic assumption that the application of thes procedures of reducing water abstraction from water sources as well as reducing the amount of wastewater produced will continue to be one of the important pillars for sustainable use of water resources. In my opinion, it would be useful, and it is also my recommendation to the authors of this work, to continue processing this set of papers focusing on an overview of the costs of unit processes, operations and entire wastewater treatment technologies for reuse. Such a study would be an important tool in the selection of procedures for the regeneration of wastewater for its reuse as well as for the effective management of water resources.

Author Response

We would like to begin by thanking the reviewer for all his comments, which have made it possible to improve the manuscript significantly. All changes are made in red. We will continue in this line, following the recommendations of the reviewer, researching this topic. In fact, we have other publications that reflect the importance of research in this line (these articles are cited in the manuscript):

  1. Gallego-Valero, L.; Moral-Pajares, E.; Román-Sánchez, I. M.; Sánchez-Pérez, J. A. Analysis of environmental taxes to finance wastewater treatment in Spain: an opportunity for regeneration? Water 2018, 10(2), 226.
  2. Moral-Pajares, E.; Gallego-Valero, L.; Román-Sánchez, I. M. Cost of urban wastewater treatment and ecotaxes: Evidence from municipalities in southern Europe. Water 2019, 11(3), 423.

Reviewer 2 Report

The article entitled "Wastewater Treatment Costs: a Research Overview through 2 Bibliometric Analysis" presents an otherwise interesting overview of the number of studies by individual countries and areas. However, there is not enough content data for an article in this category. The authors could list the basic research areas and basic findings, which is essential when writing review articles.

Author Response

We would like to begin by thanking the reviewer for all his comments, which have made it possible to improve the manuscript significantly. All changes are made in red. We have added some more information and changed last part (Discussion and Conclusions) accordingly with the reviewer’s recommendation.

We have other articles with a lot more data about this topic, and we will continue in this line, following the recommendations of the reviewer, researching this topic. In fact, we have other publications that reflect the importance of research in this line (these articles are cited in the manuscript):

  1. Gallego-Valero, L.; Moral-Pajares, E.; Román-Sánchez, I. M.; Sánchez-Pérez, J. A. Analysis of environmental taxes to finance wastewater treatment in Spain: an opportunity for regeneration? Water 2018, 10(2), 226.
  2. Moral-Pajares, E.; Gallego-Valero, L.; Román-Sánchez, I. M. Cost of urban wastewater treatment and ecotaxes: Evidence from municipalities in southern Europe. Water 2019, 11(3), 423.

Reviewer 3 Report

The title is clear.

The manuscript adheres to the journal's standards after minor revision.

The size of the article is appropriate to the contents.

The authors underline the major findings of their work and explain how the use of their proposed procedures represents a progress to other similar studies.

The Abstract is OK.

The key words permit found article in the current registers or indexes.

Please define abbreviations before usage, for example: The UN considers clean water and sanitation

In the introduction it isn’t clearly described the state of the art of the investigated problem. More references from the last years are necessary.

The methods are well described.

The paper was written in standard, grammatically correct English.

The figures have a good quality.

The table contain necessary results.

The Conclusion must be revised. Please don’t put citation in Conclusion. The Conclusion must contain major finding of this study.

References from last years are necessary.

Please provide minimum 2 references from this journal (last years), for demonstrated that manuscript is in Sustainability topics.

The paper has the text presented and arranged clearly and concisely.

Please respect guide for authors, especially in References.

Author Response

We would like to begin by thanking the reviewer for all his comments, which have made it possible to improve the manuscript significantly. All changes are made in red. We have: changed the abbreviations, improved introduction, the state of the art (adding a lot of recent references, some of them are from Sustainability), changed last part (Discussion and Conclusion) in line with the reviewer’s recommendation and checked references.

Reviewer 4 Report

This article is very well done, well structured and answers the journal's theme.

However I advise the author to rectify the comments that I presented throughout the article (Rephrase alphabetically the keywords), then to update them.

Author Response

We would like to begin by thanking the reviewer for all his comments, which have made it possible to improve the manuscript significantly. All changes are made in red. We have changed the keywords to alphabetic order.

Round 2

Reviewer 2 Report

After the amendments, the article is acceptable.